# Differences in the Incidence of Adverse Events in Acute Care Hospitals: Results of a Multicentre Study

**DOI:** 10.3390/ijerph19095238

**Published:** 2022-04-26

**Authors:** Darja Jarošová, Renáta Zeleníková, Ilona Plevová, Eva Mynaříková, Miroslava Kachlová

**Affiliations:** 1Department of Nursing and Midwifery, Faculty of Medicine, University of Ostrava, 701 03 Ostrava, Czech Republic; darja.jarosova@osu.cz (D.J.); ilona.plevova@osu.cz (I.P.); miroslava.kachlova@osu.cz (M.K.); 2Department of Nursing Care, University Hospital Ostrava, 708 00 Ostrava, Czech Republic; eva.mynarikova@fno.cz

**Keywords:** adverse events, healthcare acquired infections, safety management, multicentre study

## Abstract

Background: Adverse events are indicators of patient safety and quality of care. Adverse events clearly have negative impacts on healthcare system costs. Organizational and unit characteristics are not very often studied in relation to adverse events. The aim of the study was to find the differences in the incidence of adverse events and healthcare-associated infections in hospitalized patients in Czech acute care hospitals according to type of hospital and type of unit. Methods: This cross-sectional multicentre study was conducted in 105 acute care medical and surgical units located in 14 acute care hospitals throughout the Czech Republic. The data on adverse events and healthcare-associated infections were reported monthly by nurse researchers. The data were collected from June 2020 to October 2020. Results: The incidence of healthcare-associated infections, pressure ulcers, and medication errors was significantly lower in large hospitals. Statistically significant differences have been further found between the incidence of pressure ulcers (<0.001), falls without injury (<0.001), and falls with injury (<0.001) in surgical and medical units. More pressure ulcers, falls without injury, and falls with injury have been reported in surgical units. Conclusion: The type of hospital and type of unit affected the incidence of adverse events at acute care hospitals. To reduce adverse events, a systematic adverse event measurement and reporting system should be promoted.

## 1. Introduction

Adverse events (AEs) are indicators of quality of care and patient safety. The World Health Organization (WHO) defines an adverse event as “an injury related to medical management (as all aspects of care, including diagnosis and treatment, failure to diagnose or treat, and the systems and equipment used to deliver care), in contrast to complications of disease” [1]. Each year, around 400,000 hospitalized patients are affected by preventable error and around 100,000 people die as a result of medical error [2]. AEs obviously negatively affect the healthcare system costs as well as patients’ quality of life [3]. AEs may have a significant and serious impact on quality of care, patient satisfaction, patient safety, and length of stay [4].

AEs appear at all levels of the healthcare system. AEs can be found in different healthcare systems or disciplines, and they result from various failures [5]. The occurrence of AE events has been studied in critically ill elderly patients [6], hospitalized adult patients [7], and child patients [8]. During the last two decades, a number of papers on the incidence of AE in the healthcare field have been published in different countries (the US, Australia, the UK, Denmark). 

It is well known that AEs are multiple system failures, rather than being the individual’s fault [5]. Organizational and unit characteristics, however, are not very often studied in relation to AEs. Previous research has claimed that the type of unit and type of hospital could be among the factors associated with AEs [9,10]. The results of a Spanish study found that admission to the medical ward was an independent factor associated with AEs [9]. In a Canadian study, there was a trend for AEs to occur more frequently in university hospitals than in the broader community or in small hospitals [10].

In addition, the incidence of AEs varied across countries depending on the factors influencing the process of reporting. Nurses involved in the direct care of hospitalized patients play a key role in detecting and preventing AEs [11]. The other important role of nurses is to report AEs. Reporting all these events has a critical position in identifying and analyzing the cause of errors. The correct and consistent reporting of the errors may help develop strategies to reduce any further failures [12]. According Blegen [13], acute care units are connected with the most visible risks to patient safety. 

There are increasing numbers of integrative reviews and systematic reviews concerning AEs from different aspects [14,15,16,17,18]. Our study was focused on nurses reporting AEs in hospitalized adult patients in the Czech Republic and on finding out differences in the incidence of AEs in relation to the type of unit and hospital size. The aim of the study was to find differences in the incidence of AEs and healthcare-associated infections in hospitalized patients in Czech acute care hospitals according to type of hospital and type of unit.

## 2. Materials and Methods

### 2.1. Design

This study was a cross-sectional multicenter study conducted in 105 units located in 14 hospitals throughout the Czech Republic.

### 2.2. Sample

The study was conducted in 105 medical and surgical units located in 14 acute care Czech hospitals. Hospitals from all regions of the Czech Republic were included. Those hospitals that agreed were included in the study. Medical units were considered those providing care to acutely ill adult patients with treatment not involving surgical procedures. Surgical units were considered those providing care to adult patients requiring surgical procedures. The total number of medical and surgical units included in the study was 105.

### 2.3. Data Collection

The data were collected from June 2020 to October 2020. The total number of hospitalized patients at all 105 hospital units during the period of data collection was 1784. For the purpose of this study, the protocol was developed. The data were reported monthly by nurse research assistants to protocol. All nurse research assistants were experienced clinicians and were informed by the main researcher prior to data collection. The protocol included information about organizational and unit characteristics, AEs, and healthcare-associated infections (HAIs). The data collected included hospital size, type of unit, number of general nurses and number of practical nurses at the unit, nurses and practical nurses with absenteeism, new nurses and practical nurses at the unit, nurses and practical nurses who resigned, nursing aids at the unit, number of hospitalized patients per month, number of patient admission per month, and number of patient discharge per month. General nurses in the Czech Republic are the equivalent of registered nurses. They must have completed three years of a bachelor’s degree at a university or a diploma at a higher vocational school. Practical nurses must complete four years at a secondary nursing school. In contrast to general nurses, they have less competencies after completing their education.

AEs and healthcare-associated infections were reported monthly by nurses based on direct observation. Nurses identified AEs and HAIs as they occurred. The authors had no direct access to the medical records. 

AE is an injury related to medical management (as all aspects of care, including diagnosis and treatment, failure to diagnose or treat, and the systems and equipment used to deliver care) in contrast to the complications of disease.

Healthcare-associated infections are infections that occur in association with interactions with hospital, outpatient, or follow-up care, or that develop after discharge from a healthcare facility, and are not present or incubating at the time of admission.

From the AEs, for the purpose of this study, the following were reported monthly on each unit: falls without injury, falls with injury, pressure ulcers (newly developed), and medication errors. From HAIs, urinary tract infections, surgical site infections, gastrointestinal tract infections, peripheral line-associated bloodstream infections, skin infections, pneumonia, and respiratory tract infections were reported.

### 2.4. Data Analysis

Descriptive statistics (mean, standard deviation) were used to describe unit characteristics. The mean occurrence of AEs and HAIs was calculated per 100 patients per month. The two-sample Wilcoxon rank-sum (Mann–Whitney) test was used for analyzing AEs according to unit type and hospital type. The data were analyzed by the statistical program Stata 14. The significance level used was 5%.

Ethical committee approval was obtained before initiating the study. Additionally, approval was obtained from the nursing care hospital’s management office.

## 3. Results

From all analyzed protocols (*n* = 651), the majority of protocols were from medical units (61.75 %) and from medium-size hospitals (69.74 %). In 620 protocols, AEs and HAIs were reported.

The mean number of general nurses at the unit was 8.07, the mean number of practical nurses at the unit was 2.81, and the mean number of nursing aids at the unit was 5.29. Other organizational and unit characteristics are described in Table 1.

According to the type of hospital unit (Table 2), there was a statistically significant difference in patient-to-nurse ratio (*p* < 0.0001). The higher patient-to-nurse ratio was at surgical units, but patient-to-nurse ratio did not significantly differ based on hospital size (*p* = 0.4558) even though the patient-to-nurse ratio was higher in medium-sized hospitals. The number of nursing staff per unit was statistically significantly higher at surgical units (*p* = 0.0228). According to the hospital size, there was not a statistically significant difference in number of nurses per unit (*p* = 0.8535).

In Table 3, the mean reported incidence of AEs and healthcare-associated infections per month during the study period (5 months) is presented. The most often reported AEs were falls without injury (in 271 protocols), then pressure ulcers (in 245 protocols), followed by HAIs. From HAIs, the most frequent were urinary tract infections, followed by surgical site infections (Table 3).

Statistically significant differences have been found between the prevalence of pressure ulcers per 100 patients per month (<0.001), falls without injury (<0.001), and falls with injury (<0.001) in surgical and medical units. More pressure ulcers, falls without injury, and falls with injury have been reported in medical units (Table 4).

The prevalence of HAIs/100 patients per month (*p* = 0.0044), pressure ulcers (*p* = 0.0001), and medication errors (*p* = 0.031) was significantly lower in large hospitals (Table 5).

## 4. Discussion

The main aim of this study was to find differences in the incidence of AEs according to the type of hospital and type of unit. The number of new AEs or HAIs at each unit per month was expressed per 100 hospitalized patients. The three most often reported AEs (pressure ulcers, falls without injury, and falls with injury) have been reported more often in medical units. Similarly, more adverse drug events in nonsurgical patients have been reported in a systematic review [19]. Even though in our study the patient-to-nurse ratio was statistically significantly lower in medical units compared to surgical units, the number of all nursing staff per unit was statistically significantly higher at surgical units. To have more AEs at units where the patient-to-nurse ratio is lower is an unexpected result of our study. Using a traditional method of assignment, such as patient-to-nurse ratio, without other data (i.e., patient medical status, psychosocial status of patient, patient needs and abilities, nursing care plans) may cause an unbalanced nurse workload [20] and consequently may contribute to a higher rate of AEs, as in our research. Patients in medical units may have more comorbidities and a longer length of hospitalization, and all of these factors may lead to the development of AEs more often. The results of a Canadian study [3] suggested that one out of seven hospitalized patients at medical units experience at least one nursing-related AE. This is a significant number in terms of healthcare costs and quality of life. According to the results of a systematic review [17], one out of ten hospitalized patients is affected by AE.

The authors who developed the Pressure Ulcer and Fall Rate Quality Composite Index (100 − PUR − FR, where PUR is the pressure ulcer rate and FR is the total fall rate), which combines the hospital-acquired pressure ulcer rate and the fall rate, found that more nurses with a bachelor’s degree and higher, as well as a higher registered nurse skill mix, was associated with a higher index score [21]. Higher index scores indicated better quality, and, according to the authors [21], the index may be useful for nursing managers as a quick view of unit level nursing quality. 

In that study, the Pressure Ulcer and Fall Rate Quality Composite Index was higher in surgical units than in medical units, which means that more pressure ulcers and falls were reported at medical units [21]. This finding conforms with our results. Managers need information about their units to analyze the cause of AEs and consequently find out how to strengthen the role of nursing in patient safety. One of the results of a Croatian study [22] was that nurses with university degrees reported AEs more often. The authors of the same study [22] also stated that during exhausting times, nurses must work faster and spend less time on procedures. Shortening the nursing procedure may lead to lower quality of care. This is not too different from the phenomenon of missed nursing care described in the Missed Nursing Care Model, a new middle-range explanatory theory [23]. Missed nursing care can threaten patient safety and may potentially impact AEs. In a recent Czech study [24], there was more missed nursing care at medical units than at surgical units.

Additionally, our study observed that the incidence of HAIs, pressure ulcers, and medication errors was significantly lower in large hospitals. Larger hospitals in our sample were usually university hospitals, probably with better staffing and equipment. However, more research with a detailed analysis of organizational factors is needed to confirm this finding.

The results of an integrated review showed that the incidence of AEs in 13 selected studies ranged from 5.7–14.2%, and their preventability ranged from 31–83% [25]. The authors explained the variability by differences between hospitals, quality of care, quality of reviewer, screening criteria, sample size, and assessing the preventability rate. Understanding the factors contributing to the incidence of AEs and their reporting by staff can contribute to the analysis of the institutional policies and the improvement of management, which can increase patient safety.

### Limitation

The presented study may have several limitations. Although our study included hospitals from all districts of the Czech Republic, the selection was not random but based on the agreement of included hospitals. Some of the hospitals did not agree to participate in our research. Therefore, heterogeneity is obvious. Another limitation of the current study is a potential variability between nurse research assistants in reporting AEs and HAIs. Even though the instruction for all were the same in some units, nurse research assistants may underreport the AEs. The other limitation of our study is that more medical units than surgical units were involved in the study.

## 5. Conclusions

The type of hospital and type of unit affected the prevalence of AEs at acute care hospitals in our study. Patients from medical units in our sample seem more vulnerable, and among these patients, preventing AEs is important. To reduce AEs, a systematic AEs measurement and reporting system should be promoted. In addition, determining non-medical factors, such as psychosocial components, should be further studied in order to design multicomponent interventions for reducing AEs. 

## Figures and Tables

**Table 1 ijerph-19-05238-t001:** Organizational and unit characteristics based on analyzed protocols (*n* = 651 protocols).

Organizational Characteristics		*n*	%
Hospital size	Large	197	30.26
	Medium	454	69.74
Type of unit	Surgical	249	38.25
	Medical	402	61.75
Period of data collection	June 2020	132	20.28
	July 2021	131	20.12
	August 2021	135	20.74
	September 2021	130	19.97
	October 2021	123	18.89
**Unit characteristics**		**mean**	**SD**
General nurses at unit		8.07	3.12
General nurses with absenteeism		0.90	0.98
New general nurses at unit		0.27	0.50
General nurses who resigned		0.12	0.34
Practical nurses at unit		2.81	1.99
Practical nurses with absenteeism		0.32	0.55
New practical nurses		0.27	0.51
Practical nurses who resigned		0.12	0.34
Nursing aids at unit		5.29	3.03
Nursing aids with absenteeism		0.88	0.95
New nursing aids at unit		0.35	0.56
Nursing aids who resigned		0.22	0.47
Number of hospitalized patients per month		98.76	64.45
Number of patient admission per month		93.22	57.92
Number of patient discharge per month		92.36	57.20

**Table 2 ijerph-19-05238-t002:** Differences in organizational characteristics according to the type of unit and hospital size.

Organizational Characteristics			Mean	SD	*p*
Patient-to-nurse ratio	Type of unit	Surgical	13.00	10.81	<0.0001
		Medical	7.46	4.95	
	Hospital size	Large	8.62	4.49	0.4558
		Medium	10.00	9.31	
Number of nursing staff * per unit	Type of unit	Surgical	10.65	4.21	0.0228
		Medical	10.03	3.17	
	Hospital size	Large	10.12	2.54	0.8535
		Medium	10.33	3.99	

* General nurses (the equivalent of registered nurses) and practical nurses.

**Table 3 ijerph-19-05238-t003:** Mean occurrence of adverse events (AEs) and healthcare-associated infections (HAIs) per 100 hospitalized patients per month.

AEs and HAIs	*n*	Mean	SD	Min	Max
Falls without injury	271	1.81	1.08	1	7
Pressure ulcers (newly developed)	245	2.27	1.72	0	10
Falls with injury	117	1.49	0.98	1	6
Medication errors	11	2.54	2.42	1	8
HAIs total	188	3.38	4.23	0	28
Urinary tract infections	145	2.12	1.73	1	10
Surgical site infections	72	2.08	1.93	1	13
Gastrointestinal tract infections	68	1.69	1.17	1	5
Peripheral line-associated bloodstream infections	47	1.76	1.54	1	7
Skin infections	46	1.28	0.50	1	3
Pneumonia	45	1.73	1.46	1	10
Respiratory tract infections	32	1.43	0.76	1	4

**Table 4 ijerph-19-05238-t004:** Differences in incidence of AEs (*n*/100 patients) per month based on unit type (*n* = 620 protocols).

Variable	Type of Unit	*n*	Mean	SD	Min	Max	*p* *
Healthcare acquired infections	Surgical	241	0.89	2.36	0	16.5	0.1792
	Medical	379	2.31	7.26	0	75	
Pressure ulcers	Surgical	241	0.45	0.85	0	4.9	<0.001
	Medical	379	2.05	4.30	0	31.25	
Falls without injury	Surgical	241	0.37	0.68	0	4.4	<0.001
	Medical	379	1.81	3.73	0	50	
Falls with injury	Surgical	241	0.14	0.57	0	5.1	<0.001
	Medical	379	0.64	1.89	0	20	
Medication errors	Surgical	241	0.00	0.04	0	0.6	0.0582
	Medical	379	0.17	1.82	0	32	

* Two-sample Wilcoxon rank-sum (Mann–Whitney) test.

**Table 5 ijerph-19-05238-t005:** Differences in the prevalence of AEs (*n*/100 patients) per month based on hospital size (*n* = 620 protocols).

Variable	Hospital Size	*n*	Mean	SD	Min	Max	*p* *
Healthcare acquired infections	Large	195	0.70	2.56	0	29.17	0.0044
	Medium	425	2.24	6.86	0	75	
Pressure ulcers	Large	195	1.37	3.97	0	31.25	0.0001
	Medium	425	1.46	3.25	0	30	
Falls without injury	Large	195	1.45	4.22	0	50	0.946
	Medium	425	1.16	2.29	0	20	
Falls with injury	Large	195	0.33	1.25	0	10	0.1154
	Medium	425	0.50	1.66	0	20	
Medication errors	Large	195	0.00	0	0	0	0.031
	Medium	425	0.16	1.72	0	32	

* Two-sample Wilcoxon rank-sum (Mann–Whitney) test.

## Data Availability

The data presented in this study are available on request from the corresponding author.

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
