# Peer review of "Differences in the Incidence of Adverse Events in Acute Care Hospitals: Results of a Multicentre Study"

_ijerph, 2022, doi:10.3390/ijerph19095238_

Round 1

Reviewer 1 Report

General Comments:

This paper investigated the differences in the incidence of adverse events and healthcare-associated infections in hospitalized patients in acute care hospitals based on a cross-sectional multicentre study with 14 acute care hospitals throughout the Czech Republic. The main conclusion is that type of hospital and type of unit are significantly associated with the incidence of adverse events at acute care hospitals.

Research background and related concepts in this paper are not very clear. Besides, this manuscript is not well written, containing numerous gramma errors and typos, needs improvement. There are several specific issues in the paper need to be addressed. Please see the specific comments as follows.

Specific Comments:

  1. In the Abstract, Background, the claim “Rates of adverse events are high, …” is inaccurate. The incidences of many adverse events (AEs) can be very low in practice, depending on the level of AEs (grade 3 or higher), disease type, treatment, dose-level, and many other factors. Besides, in the Introduction, the authors also reviewed that “The overall average of the incidence of AEs in Italian study was 5.2%, the incidence of AE in Danish study was 9% of all admissions….”. The concept of high or low is relative, not absolute.
  2. Does this research put emphasis on the treatment related AEs, or healthcare related AEs during hospitalized stay (such as the healthcare-associated infections, HAIs), or all kind of AEs? The background or concept of AEs is not quite clear. From Table 3, it seems that the AEs are mainly healthcare related.
  3. Section 2.1, the description of the study design is too brief, containing even no single complete sentence with full stop.
  4. Page 2, row 77, it is said that “the total number of hospitalized patients at all hospital units during the period of data collection was 1,784”. But, in the section of Results, page 3, row 103, the authors only included 651 patients in the study. How did you select the patients? What is the selection criterion? On page 4, why the number reduces to n = 620 in Table 4 ?
  5. This paper showed the existence of potential correlation between the type of hospital (or type of unit) and the incidence of adverse events at acute care hospitals using non-parametric statistical hypothesis test (Mann-Whitney test). It is recommended to consider some direct statistical models to quantify the association between the types of hospital and unit and the incidence of AEs at acute care hospitals.
  6. Utilizing figures or other visualization tool to present or summarize the results is highly recommended.
  7. How did the authors account for the heterogeneity of AEs in multicentre study?

Minor issues:

  1. Page 1, row 33, please double check “AEe events”.
  2. Page 1, row 41, gramma issue or typo “AE are preventable”.
  3. No need to center the ‘Conflicts of Interest’.

Author Response

Specific Comments:

  1. In the Abstract, Background, the claim “Rates of adverse events are high, …” is inaccurate. The incidences of many adverse events (AEs) can be very low in practice, depending on the level of AEs (grade 3 or higher), disease type, treatment, dose-level, and many other factors. Besides, in the Introduction, the authors also reviewed that “The overall average of the incidence of AEs in Italian study was 5.2%, the incidence of AE in Danish study was 9% of all admissions….”. The concept of high or low is relative, not absolute.

This was rewritten. Information about the incidence was deleted.

  1. Does this research put emphasis on the treatment related AEs, or healthcare related AEs during hospitalized stay (such as the healthcare-associated infections, HAIs), or all kind of AEs? The background or concept of AEs is not quite clear. From Table 3, it seems that the AEs are mainly healthcare related.

The research focused on healthcare related AEs during hospital stay.

  1. Section 2.1, the description of the study design is too brief, containing even no single complete sentence with full stop.

Te description of the study design was rewritten to complete sentence.

  1. Page 2, row 77, it is said that “the total number of hospitalized patients at all hospital units during the period of data collection was 1,784”. But, in the section of Results, page 3, row 103, the authors only included 651 patients in the study. How did you select the patients? What is the selection criterion? On page 4, why the number reduces to n = 620 in Table 4 ?

The number 651 is the number of research protocols with nurses monthly reports of AEs and HAIs, and number of 620 is the number of protocols with reported AEs.

  1. This paper showed the existence of potential correlation between the type of hospital (or type of unit) and the incidence of adverse events at acute care hospitals using non-parametric statistical hypothesis test (Mann-Whitney test). It is recommended to consider some direct statistical models to quantify the association between the types of hospital and unit and the incidence of AEs at acute care hospitals.

In submitted paper statistical models will  not be presented.

  1. Utilizing figures or other visualization tool to present or summarize the results is highly recommended.

Due to limited number of tables and figures no other visualizaton tool was added.

How did the authors account for the heterogeneity of AEs in multicentre study? The frailty model was not calculated.

Reviewer 2 Report

The paper presents the important issue of adverse events. It presents interesting data, but I have several concerns and suggestions for the authors:

Abstract: should be reorganized. Add N to results and the number of AE for each type of incidence. Conclusions are too general and weak.

Introduction: should be significantly improved. The first paragraph is too long, presenting mainly numbers that are not directly connected to your study. Also, the section on reporting and underreporting is not well related to the aim of your research and especially to the variables you examined: type of medical unit and hospital size as variables affecting AE rates, and types of AE: falls, medication errors, and HAIs. Please add some literature on these factors and types of AEs.

Method: how many acute care hospitals did you apply to in the Czech Republic? It is confusing to the reader that you mentioned including 14 acute care hospitals, and later you mentioned including 105 acute care wards. Please clarify.

You have mentioned that "AEs and healthcare-associated infections were reported monthly based on patient records and nursing documentation". Please clarify how you decided to define AE for the study's purpose and the difference between patient records and nursing documentation.  

Results: In table 1: the N, mean, and SD of unit characteristics are confusing. Please clarify. The rationale for describing the differences between practical nurses and general nurses in the results section is not clear. You did not present any data showing differences between the type of nurses in AE reporting rates, so it is not fully understood.

Discussion: this section is not well connected to your results (and not related to your intro). You talk about nurses' skills and degrees even though you didn’t present any analysis of this variable in your results section. Then you talk about the organizational culture, which may be an important, influential factor, but you didn’t discuss it in the introduction section, nor did you present the organizational culture in the hospitals you examined. Conclusions should be significantly improved; they are very weak and not entirely related to the findings in terms of implications. Add some recommendations.

Author Response

Abstract: should be reorganized. Add N to results and the number of AE for each type of incidence. Conclusions are too general and weak.

Introduction: should be significantly improved. The first paragraph is too long, presenting mainly numbers that are not directly connected to your study.

The first paragraph was shortened as recomended, numbers were deleted.

Also, the section on reporting and underreporting is not well related to the aim of your research and especially to the variables you examined: type of medical unit and hospital size as variables affecting AE rates, and types of AE: falls, medication errors, and HAIs. Please add some literature on these factors and types of AEs.

Section on reporting was deleted from introduction. Some literature about type of hospital and type of unit was added.

Method: how many acute care hospitals did you apply to in the Czech Republic? It is confusing to the reader that you mentioned including 14 acute care hospitals, and later you mentioned including 105 acute care wards. Please clarify.

This was clarified as recommended. The sample included 105 acute care wards from 14 acute care hospitals.

You have mentioned that "AEs and healthcare-associated infections were reported monthly based on patient records and nursing documentation". Please clarify how you decided to define AE for the study's purpose and the difference between patient records and nursing documentation.  

The statement about patient records and nursing documentation was revised. It was misinterpreted – this was meant that nursing observations of AEs were recorded into patient medical record or separate nursing process documentation and subsequently information was transfered to protocol. But for readers it is more clear to state that AEs and HAIs were reported based on observation.

Definitions of AEs and HAIs were added to Data collection part.

Results: In table 1: the N, mean, and SD of unit characteristics are confusing. Please clarify. The rationale for describing the differences between practical nurses and general nurses in the results section is not clear. You did not present any data showing differences between the type of nurses in AE reporting rates, so it is not fully understood.

Confusing N in Table 1 was deleted. The description of the differences between practical nurses and general nurses was moved to data colletion section. The rationale is to explain the readers how these

Discussion: this section is not well connected to your results (and not related to your intro). You talk about nurses' skills and degrees even though you didn’t present any analysis of this variable in your results section. Then you talk about the organizational culture, which may be an important, influential factor, but you didn’t discuss it in the introduction section, nor did you present the organizational culture in the hospitals you examined. Conclusions should be significantly improved; they are very weak and not entirely related to the findings in terms of implications. Add some recommendations.

Discussion and conclusion were revised as recommended.

Round 2

Reviewer 1 Report

I am sorry to see that this revision is not satisfactory.    Over two fundamental concerns  were not addressed properly. Major revision is necessary.

Author Response

I am sorry to see that this revision is not satisfactory.  Over two fundamental concerns  were not addressed properly. Major revision is necessary.

Response: In order to explain all previous comments we response to all again and in more detail. We would like to apologize for missed response to some of the reviewer comments which was caused probably by incorrect copying from separated document. Now we double checked that all comments are properly addressed if applicable and possible.

Thank you very much for your understanding.

General Comments:

This paper investigated the differences in the incidence of adverse events and healthcare-associated infections in hospitalized patients in acute care hospitals based on a cross-sectional multicentre study with 14 acute care hospitals throughout the Czech Republic. The main conclusion is that type of hospital and type of unit are significantly associated with the incidence of adverse events at acute care hospitals.

Research background and related concepts in this paper are not very clear. Besides, this manuscript is not well written, containing numerous gramma errors and typos, needs improvement. There are several specific issues in the paper need to be addressed. Please see the specific comments as follows.

Response: Thank you reviewer for all comments which we tried to address.  

Specific Comments:

In the Abstract, Background, the claim “Rates of adverse events are high, …” is inaccurate. The incidences of many adverse events (AEs) can be very low in practice, depending on the level of AEs (grade 3 or higher), disease type, treatment, dose-level, and many other factors. Besides, in the Introduction, the authors also reviewed that “The overall average of the incidence of AEs in Italian study was 5.2%, the incidence of AE in Danish study was 9% of all admissions….”. The concept of high or low is relative, not absolute.

Response: Thank you for this comment. Background and abstract was rewritten. Information about the incidence was deleted. Instead, one paragraph on organizational and unit characteristics as potential factors associated with AEs, was added.

Does this research put emphasis on the treatment related AEs, or healthcare related AEs during hospitalized stay (such as the healthcare-associated infections, HAIs), or all kind of AEs? The background or concept of AEs is not quite clear. From Table 3, it seems that the AEs are mainly healthcare related.

Response: The research focused on healthcare related AEs during hospital stay. To be more clear for readers, definition of AEs and HAIs was added to Data collection section.

Section 2.1, the description of the study design is too brief, containing even no single complete sentence with full stop.

Response: Te description of the study design was rewritten to complete sentence.

Page 2, row 77, it is said that “the total number of hospitalized patients at all hospital units during the period of data collection was 1,784”. But, in the section of Results, page 3, row 103, the authors only included 651 patients in the study. How did you select the patients? What is the selection criterion? On page 4, why the number reduces to n = 620 in Table 4 ?

Response: The total number of hospitalized patients at all 105 hospital units during the period of data collection was 1,784. Not all these patients experienced AEs or HAIs. Based on direct observation nurses monthly reported AEs and HAIs as they occur. The number 651 is the number of research protocols with nurses monthly reports of AEs and HAIs, and number of 620 is the number of protocols with reported AEs. For statistical purposes the mean occurrence of AEs and HAIs was calculated per 100 patients per month.

This paper showed the existence of potential correlation between the type of hospital (or type of unit) and the incidence of adverse events at acute care hospitals using non-parametric statistical hypothesis test (Mann-Whitney test). It is recommended to consider some direct statistical models to quantify the association between the types of hospital and unit and the incidence of AEs at acute care hospitals.

Response: To quantify the association between the types of hospital and unit and the incidence of AEs at acute care hospitals two-sample Wilcoxon rank-sum (Mann–Whitney) test was used. We do not have previous input data from hospitals to run some direct statistical models. Our research was focused differently which is presented in submitted paper.

Utilizing figures or other visualization tool to present or summarize the results is highly recommended.

Response: Due to limited number of tables and figures no other visualizaton tool was added. We consider data presented in tables as sufficient.

How did the authors account for the heterogeneity of AEs in multicentre study?

Response: Although our study included hospitals from all districts of the Czech Republic, the selection was not random but based on the agreement of included hospitals. Some of the hospitals did not agree to participate in our research. Therefore, heterogeneity is obvious. The strategy for addressing heterogeneity of AEs in our study was checking that the data are correct. Organizational and unit characteristics were presented in tables.

Minor issues:

  1. Page 1, row 33, please double check “AEe events”.
  2. Page 1, row 41, gramma issue or typo “AE are preventable”.
  3. No need to center the ‘Conflicts of Interest’.

Response: Native English speaker and English language professional revised the paper and corrected the gramma errors and typos.

Reviewer 2 Report

I want to thank the authors for addressing my comments. 

Author Response

Thank you reviewer for positive comments.